# Novel Genetic Loci from *Triticum timopheevii* Associated with Gluten Content Revealed by GWAS in Wheat Breeding Lines

**DOI:** 10.3390/ijms241713304

**Published:** 2023-08-27

**Authors:** Irina N. Leonova, Antonina A. Kiseleva, Alina A. Berezhnaya, Olga A. Orlovskaya, Elena A. Salina

**Affiliations:** 1The Federal Research Center Institute of Cytology and Genetics SB RAS, Novosibirsk 630090, Russia; antkiseleva@bionet.nsc.ru (A.A.K.); berezhnaya@bionet.nsc.ru (A.A.B.); salina@bionet.nsc.ru (E.A.S.); 2Kurchatov Genomics Center IC&G SB RAS, Novosibirsk 630090, Russia; 3Institute of Genetics and Cytology of the National Academy of Sciences of Belarus, 220072 Minsk, Belarus; o.orlovskaya@igc.by

**Keywords:** wheat, gluten content, alien introgressions, association mapping, *Triticum timopheevii*

## Abstract

The content and quality of gluten in wheat grain is a distinctive characteristic that determines the final properties of wheat flour. In this study, a genome-wide association study (GWAS) was performed on a wheat panel consisting of bread wheat varieties and the introgression lines (ILs) obtained via hybridization with tetraploid wheat relatives. A total of 17 stable quantitative trait nucleotides (QTNs) located on chromosomes 1D, 2A, 2B, 3D, 5A, 6A, 7B, and 7D that explained up to 21% of the phenotypic variation were identified. Among them, the QTLs on chromosomes 2A and 7B were found to contain three and six linked SNP markers, respectively. Comparative analysis of wheat genotypes according to the composition of haplotypes for the three closely linked SNPs of chromosome 2A indicated that haplotype TT/AA/GG was characteristic of ten ILs containing introgressions from *T. timopheevii*. The gluten content in the plants with TT/AA/GG haplotype was significantly higher than in the varieties with haplotype GG/GG/AA. Having compared the newly obtained data with the previously reported quantitative trait loci (QTLs) we inferred that the locus on chromosome 2A inherited from *T. timopheevii* is potentially novel. The introgression lines containing the new locus can be used as sources of genetic factors to improve the quality traits of bread wheat.

## 1. Introduction

Wheat (*Triticum aestivum* L.) is one of the most important food crops for the world’s population. Consuming a variety of products obtained from processed wheat grains, humanity receives up to 20% of calories and protein, as well as micro- and macroelements and vitamins [1]. However, despite increasing wheat production, all world producers suffer from the low quality of food grain [2,3,4,5].

Protein content in grain is considered a crucial quality trait for determining its nutritional value. Gliadin and glutenin, storage proteins of wheat, are principal constituents of wet gluten and significantly influence the viscoelasticity and baking quality of wheat dough [6,7], so the content and composition of these proteins are of great importance in evaluating wheat quality [8]. Gliadins are responsible for the viscosity and extensibility of the dough, and glutenins for strength and elasticity, while the balance of these proteins affects the quality of the flour [9,10]. At present, a large amount of data has been accumulated on the structure of storage protein complexes, their allelic composition, and the polymorphism of genes encoding gliadins and glutenins. In particular, it has been established that the main gliadin-coding loci are located in the short arms of the 1st (*Gli-1*) and 6th (*Gli-2*) homoeologous chromosome groups [11,12]. Glutenins differ in subunit composition, while the key role in the formation of baking qualities belongs to high molecular weight glutenin subunits (HMW-GS) encoded by the *Glu-1* loci in chromosomes 1A, 1B, and 1D [13,14,15]. Each HMW-GS locus contains two genes encoding subunits of x- and y-types, and up from three or five out of six HMW-GS genes are expressed in wheat [16,17,18].

Wild and cultivated relatives of wheat have a great potential for expanding the crop’s genetic diversity in agronomically important traits. The significant allelic diversity of closely related and distant species from the secondary and tertiary gene pools for the genes and quantitative trait loci (QTLs) that increase resistance to biotic and abiotic stress factors is used to improve bread and durum wheat [19,20,21]. As for quality traits, until recently, the main attention of researchers has been paid exclusively to such species as *Triticum durum*, *T. dicoccum*, and *T. dicoccoides* since the A and B genomes of these species are evolutionarily closer to common wheat.

Gluten content (GC) and its composition vary significantly among different members of the Triticeae tribe. It has been demonstrated that with a decrease in ploidy, both the total gluten content and the diversity of the HMW-GS composition increase [22,23,24]. Some studies have proposed that the A and B genomes of durum and emmer wheat may be responsible for the majority of crucial grain quality parameters [25,26]. In the genome of wild-spelt *T. dicoccoides*, a functional allele of the *Gpc-B1* gene was found that increased the protein content in the grain. Despite the different effects the gene had depending on the recipient variety and environmental influence, the transfer of *Gpc-B1* to bread and durum wheats using MAS technologies improves the protein content of new genotypes [27,28,29]. Among tetraploid species, *Triticum dicoccoides* is also highly diverse if the 1Ay subunit is considered [22], so there have been cases of successful transfer of the *Glu-A1* alleles from *T. dicoccoides* into the genome of common wheat to increase the protein and gluten content and improve technological flour and dough properties [30,31]. Replacement of *Glu-A1* with *Glu-A^th^1* encoding subunits 39 + 40 from *T. boeoticum* Boiss. ssp. *thaoudar* has also improved dough strength. The introduction of this allele has resulted in dough stickiness reduction and enhanced its stability while mixing, so it was suggested to be potentially valuable in breeding programs aimed at improving bread-making quality [32]. Transferring the *Glu-S^s^1* and *Gli-S^s^1* loci from *Aegilops searsii* into the Chinese Spring cultivar has led to the new genotypes with improved indicators such as sodium dodecyl sulfate (SDS) sedimentation, dough strength, etc., if compared with the recipient cultivar [33].

For species from the secondary and tertiary gene pools, the genomic composition of which differs from the A, B, and D genomes of common wheat, the published data on their quality traits are scarce, and most of them are devoted mainly to investigating the allelic diversity of storage proteins and their possible impact on improving quality indicators [34,35,36]. Increased content of wet gluten and protein in the grain has been found in the hybrid lines containing alien chromatin from species of Timopheevi group (*T. timopheevii*, *T. araraticum*, genomes A^t^A^t^GG), *Aegilops speltoides* (genome S), wheatgrass *Thynopyrum intermedium* (genomes EEE^st^E^st^S^t^S^t^) [24,37,38,39,40], which makes it possible to assume that the introgression fragments carried over from wheat relatives may contain the genetic factors affecting the GC.

Using such modern molecular technologies as gene and QTL mapping has shown that in the genome of bread and durum wheat, the content of protein and gluten in grain, as well as a number of characteristics related to the quality of flour, is controlled by a large number of the loci located in almost all chromosomes [8,41,42,43]. As for the species with genomes other than A, B, and D, the data on chromosomal localization and loci mapping for the quality traits inherited from these species are currently lacking. The present study aimed to characterize the variation in gluten content and composition of HMW-GS in a wheat panel consisting of bread wheat varieties and the introgression lines obtained through hybridization with tetraploid wheat relatives and to search for the valuable loci and candidate genes associated with these traits. To map the QTLs, a genome-wide association study (GWAS) was used since this methodological approach has proved to be effective in the dissection of complex quantitative traits, such as yield and quality traits in many crop types [26,44].

## 2. Results

### 2.1. Phenotypic Assessment

According to the data from two field seasons, the average GC varied from 22.8 to 44.2%. The highest rates (>36%) were found in *T. kiharae*, three bread wheat varieties, and 14 ILs (Figure 1, Appendix A). Comparative group analysis of the varieties and ILs indicates that the GC was 1.2 times higher in the introgression lines (*p* < 0.001) (Appendix A). Differences in GC values were also noted for both varieties and ILs by growing seasons. For the varieties, there was a significant decrease in GC in 2019 compared to 2018 (27.4 vs. 30.0, *p* < 0.00001). The ILs, on the other hand, demonstrated an opposite trend, so their GC in 2019 was higher (36.5 vs. 33.1%, *p* < 0001). Broad-sense heritability estimated for GC showed the trait’s high heritability (*H*^2^ = 0.92). ANOVA data indicated that genotype, environments, and their interaction influenced the phenotypic manifestation of GC (Appendix A).

### 2.2. Allelic Diversity of HMW-GS

The allelic composition of HMW-GS was determined using previously developed PCR primers for the *Glu-A1*, *Glu-B1*, and *Glu-D1* loci (Appendix A) [45,46,47,48]. Of the three subunits of the *Glu-A1* locus, the highest occurrence frequency (58%) was noted for the Ax2 subunit compared to Ax1 and Ax0, which were found in 31 and 25 samples, respectively (Appendix A). Among the wheat cultivars, 87% of the samples contained Ax2 or Ax1 subunits. In the group of introgression lines, the high occurrence frequency of the Ax0 subunit (40%) was noted, while Ax1 was found only in one line (Appendix A).

Subunit composition analysis of the *Glu-B1* locus showed that 105 samples out of 137 had the *Glu-B1c* allele (Bx7* + By9 subunits). A combination of Bx7* + By8 subunits was found only in the Samsar variety. In a tetraploid species *T. timopheevii*, a synthetic hexaploid *T. kiharae*, and in two ILs (190/5-3 and 190/6-1), the amplification fragments differed in size from the PCR products characteristic of the Bx7 subunit of the Chinese Spring control cultivar (Appendix A). Since the ILs were obtained by crossing the Chinese Spring variety with *T. durum*, it can be assumed that the Bx subunit of these ILs was inherited from *T. durum*. The other introgression lines inherited *Glu-B1* loci from parental common wheat varieties. The subunits characteristic of *T. timopheevii* and *T. durum* were conventionally designated as Bx-Tt and Bx-Td.

Application of the used set of PCR primers to HMW-GS loci did not make it possible to determine exactly which of the three By subunits 8*, 18, or 15 was present in the genome of 26 samples (Appendix A). However, differences were noted in the PCR spectra for the species *T. dicoccoides* if compared to Chinese Spring carrying the By8 allele [45]. Therefore, based on the PCR results for By subunits detection, it was hypothesized that the *T. dicoccoides* sample contained either a By18 or a By26 subunit (Appendix A). The *T. timopheevii* and *T. kiharae* species that lacked amplification products for y-type subunit primers of the *Glu-B1* locus were designated as non-By ones.

The *Glu-D1* locus in the studied sample panel was represented by the *Glu-D1a* (Dx2 + Dy12) and *Glu-D1d* (Dx5 + Dy10) alleles, whose occurrence frequency was almost the same both within the entire collection and groups consisting of commercial varieties and ILs.

The GC was analyzed depending on the presence of different alleles of *Glu-A1*, *Glu-B1* and *Glu-D1* loci. For the *Glu-A1* locus, it was shown that the GC in the varieties containing the Ax0 subunit was higher than in the samples with other *Glu-A1* alleles (Figure 2A). For the alleles of the *Glu-B1* and *Glu-D1* loci, no significant differences in GC were detected (Figure 2B,C).

### 2.3. Population Structure and Association Study

Studying the genetic structure of a population of 137 samples using the STRUCTURE-like inference algorithm LEA [49] suggested that the number of postulated clusters was seven, including 25, 27, 15, 33, 18, 14, and 7 genotypes, respectively (Appendix A). It was noted that this grouping indicated the absence of a clear separation according to their origin from different breeding centers, so the ILs were clustered in groups with the original parental common-wheat varieties.

The GWAS performed by applying the FarmCPU algorithm [50] detected 17 significant QTNs for GC at *p*-value < 0.001 to be located on chromosomes 1D, 2A, 2B, 3D, 5A, 6A, 7B, and 7D, and explain from 0.5 to 21% of the phenotypic variation (Table 1, Figure 3). These significant associations were identified both across the two environments and based on best linear unbiased estimates (BLUEs) (Appendix A).

In chromosome 2A, QTL was located within 95,472,680–98,681,112 bp of the physical map (RefSeq v.2.1) and included three markers, namely wsnp_BE498730A_Ta_2_2, Tdurum_contig45580_1717 and Excalibur_c21269_176. On its genetic consensus map, these SNPs were mapped in the same region of 101.97 cM [51].

Six MTAs (Kukri_c55922_352, RFL_Contig801_2124, Tdurum_contig98005_345, Tdurum_contig98005_272, Excalibur_c60612_236, Tdurum_contig5352_556) were revealed on the short arm of the 7B chromosome in the 4,123,555–5,395,003 bp region (RefSeq v.2.1). According to the chromosome’s genetic map, three of them (RFL_Contig801_2124, Tdurum_contig98005_272, and Tdurum_contig5352_556) were located at 10.06 cM, as for the other three markers their localization was not defined.

Two highly significant MTAs (wsnp_Ex_c4921_8764106 and RAC875_rep_c112818_307) were found in the long arm of chromosome 5A to be located at 475,037,107 and 615,212,548 bp of the physical map and at 53.46 and 98.90 cM of the consensus genetic m3ap. The loci’s contribution to the trait’s phenotypic manifestation was 11.64 and 21.15%, respectively.

Two SNP markers (Tdurum_contig64563_491, RAC875_c14105_66, RFL_Contig3016_1091) detected on chromosome 2B determined up to 2.23% of the trait variation. The markers were located in different regions of the physical map and were not linked to each other as their LD analysis demonstrated. Also, single MTAs were identified on chromosomes 1D, 3D, 6A, and 7D.

According to the LD analysis based on paired *r*^2^ values between SNPs, the closed linkage was shown for three SNP markers at locus 2A and six SNPs at locus 7B (Figure 3C). For the SNPs at 2A, a comparative analysis of the collection for the composition of haplotypes was carried out to discover that the GG/GG/AA haplotype occurred in all commercial varieties of common wheat and ten ILs derived predominantly from *T. durum*. The TT/AA/GG haplotype was characteristic of *T. timopheevii*, *T. kiharae*, *T. dicoccum*, and ten ILs obtained from the hybridization of wheat cultivars with *T. timopheevii* (Appendix A). It was also shown that the content of gluten in plants with the GG/GG/AA haplotype was significantly lower if compared to the TT/AA/GG haplotype (Figure 4).

Based on the results of genotyping with SNP markers and previous SSR marking for ILs, we estimated the putative length of the alien fragments in chromosome 2A in the lines containing introgression from *T. timopheevii*. It turned out that in line with the GG/GG/AA haplotype, the introgressed fragment was located outside the localization area of the most significant markers wsnp_BE498730A_Ta_2_2, Tdurum_contig45580_1717 and Excalibur_c21269_176; no introgressions in 2A was detected in the three ILs (Appendix A).

We carried out a comparative analysis of the number of effective alleles of the loci associated with GC in the genomes of the wheat varieties. It turned out that in the group of samples with a low GC (22–27%), the number of loci is, on average, 5.9, while in the group with a gluten content of more than 35%, the number of loci is significantly higher (10.2) (Appendix A). A high GC (>35%) was detected in 17 varieties, including 13 ILs, *T. dicoccoides,* and *T. kiharae,* and only two varieties of bread wheat (Samsar and Rassvet). It was also shown that GC-associated loci in chromosomes 1D, 6A, and 7D are detected in most varieties of this collection, regardless of their origin and the presence of alien introgressions.

### 2.4. Putative Candidate Gene Analysis

#### 2.4.1. Locus 2A

In total, eleven genes are localized in the significant marker region (95,472,680–98,681,112 bp). To select candidate genes, the analyzed region was expanded by ~100 Kb in both directions from the extreme markers. This expanded locus included 16 genes (Appendix A). Since the genes encoding gluten proteins tend to be predominantly expressed in grain [52], the genes with a similar expression pattern were prioritized as candidates.

Gene expression evaluation showed that gene *TraesCS2A03G0302500* (RefSeq v. 1.1 *TraesCS2A02G147400*) had predominant expression in grain during hard dough and ripening stages since it encoded eukaryotic translation initiation factor 6 (TaELF6). The expression patterns of other genes from this locus were non-tissue specific and were expressed in most plant tissues with different intensities.

#### 2.4.2. Locus 7B

The locus in the short arm of chromosome 7B included six markers associated with GC (Table 1). These markers cover a 1,271,448 bp region in which 21 genes were annotated (Appendix A). The only gene whose expression was found in the grain and most strongly in the endosperm was *TraesCS7B03G0016400* (RefSeq v. 1.1 *TraesCS7B02G006800*).

#### 2.4.3. Other Loci

Of the other genes located in the area for significant MTAs (Table 1), *TraesCS5A03G0647900* (RefSeq v. 1.1 *TraesCS5A02G260600*) located on chromosome 5A was most significantly expressed in grain during hard dough and in embryo while ripening (Appendix A). *TraesCS5A03G0647900* encodes heat shock protein 90. Another gene, *TraesCS1D02G159700,* encoding the seed storage protein, was located on chromosome 1D in the region of 436 191 953 bp (RefSeq v. 1.1) of the physical map.

## 3. Discussion

### 3.1. Comparison of Putative QTL Localization with Known Data

In this study, the GWAS approach was used to map GC-associated loci in grain. Having analyzed a collection including commercial varieties of spring bread wheat and the introgression lines with the genetic material of Triticeae species, we detected 17 stable QTNs on eight chromosomes 1D, 2A, 2B, 3D, 5A, 6A, 7B, and 7D. The loci were identified with high significance based on trait assessments during two growing seasons as based on BLUEs. A comparative analysis of QTN localization was conducted to compare the obtained results against those from previous studies.

According to the published data obtained using biparental mapping population and GWAS, GC loci were found in almost all common wheat chromosomes [43,44,53,54,55,56,57]. In our study, we detected a locus in chromosome 1D located in the region of 436,191,827 bp (RefSeq v. 1.1) of the physical map. Near significant marker wsnp_Ex_c35886_43950102, the *TraesCS1D02G159700* gene encoding the seed storage protein was located, which suggests that the locus may belong to *Glu-D1*. A similar locus containing the *TraesCS1D02G159700* gene was discovered by Mohamed et al. [58] when studying the wheat lines obtained from crosses with various accessions of *Aegilops tauschii*. In another research by Deng et al. [54], two pleiotropic loci associated with GC and gluten index were identified using the DArT and SSR markers in chromosome 1D in marker interval *Glu-D1-wPt3743*. According to Alemu et al. [56], the GWAS performed using several statistical models showed the presence of stable associations with wet gluten content on chromosome 5A (region 476,947,119 Mb) across all models. The *qGC5A* locus located at 479.28–479.32 Mb was mapped in chromosome 5A in the study of Chinese common wheat varieties [57], and the position of these QTLs coincides with one of the loci on chromosome 5A detected in our study (marker wsnp_Ex_c4921_8764106). In the region of this marker is the *TraesCS5A03G0647900* gene, characterized by increased expression in grain. Previously, this gene was shown to be involved in the response to drought stress [59].

There is experimental evidence for the presence of GC-associated QTLs in different regions of chromosome 6A. Thus, using a collection of 486 bread wheat samples, including modern cultivars, landraces, and breeding lines from regions worldwide, an important QTL (*QNWGC.cau-6A*) for wet GC was found in the short arm of chromosome 6A at 73.58 Mb [60]. According to Alemu et al. [56], a region containing valuable QTLs was found in the long arm of chromosome 6A (496.84 Mb).

A number of studies have demonstrated QTL colocalization for both protein and gluten content [57,61,62]. Previously, we detected four QTLs to control protein content in the grain of the commercial varieties included in this collection [63]. It turned out that the significant loci *QGpc.icg-6A.2* responsible for GPC is located in the same region of chromosome 6A as the GC locus identified in this study. These results suggest that the 61–73 Mb region harbors the genetic factors associated with quality traits.

A stable locus was identified on chromosome 7B, which includes several closely linked markers. GC-associated markers for chromosome 7B were previously described in the literature [64], but their localization does not coincide with the locus mapped in our study. In the region of this locus, 21 genes were annotated, of which only *TraesCS7B03G0016400* (RefSeq v. 1.1 *TraesCS7B02G006800*) is expressed in grain. This gene encodes transcription elongation factor 1. Previously, *TraesCS7B02G006800* was proposed as a candidate that affects embryo length and width [65] as well as the zinc content in grain under restricted irrigation conditions [66].

Associations in the 3D chromosome (region 531,375,739 bp) with gluten content were found by Gao et al. [44]. However, the authors noted that MTA appeared only in one environment out of six. On chromosome 7D, Yang et al. [8] mapped two loci *q7D-4* and *q7D-5* (626,151,126–634,138,826 bp region), which showed associations with several quality traits, such as GC, protein content, and dough rheological properties. However, the position of these QTLs is different from the locus identified in our work.

Important data on the genetic dissection of gluten-related traits were obtained for durum wheat, including elite cultivars, landraces, breeding lines, and subspecies. Ortho-meta QTL analysis across cereal species, conducted on the basis of ten GWAS studies for varieties and species of durum wheats, revealed hot spot QTLs for grain protein content and gluten index in chromosomes 1A, 1B, 3B, 5A, 6A, and 7A [67]. In another study, Jonson et al. [25] analyzed 24 traits related to protein content, gluten strength, cooking, and milling quality, wherein major and stable QTLs for gluten-related traits were identified on chromosomes 1A and 1B.

We analyzed 16 genes in the region of the locus located on chromosome 2A (Appendix A). A grain-specific expression pattern was demonstrated only for *TraesCS2A03G0302500* (RefSeq v. 1.1. *TraesCS2A02G147400*) encoding eukaryotic translation initiation factor 6 (TaELF6). This gene had previously been shown to be involved in seed dormancy regulation [68].

The haplotype in the 2A chromosome that in our study was associated with high GC came from the wheat relatives, so we compared this region in *T. aestivum*, *T. dicoccoides*, *T. turgidum,* and *T. urartu*, the wheat species for which genome sequencing had been performed. It turned out that the region differed in its gene composition in the common wheat and wild relatives. The analysis showed that the species of *T. dicoccoides*, *T. turgidum*, and *T. urartu* contained genes not found in the genome of *T. aestivum*. For example, take gene *TRIDC2AG018400*/*TRIDT2Av1G042270*/*TUG1812G0200001533.01* encoding basic pentacysteine1 (domain of unknown function DUF1618). Its orthologue in *T. aestivum* (*TraesCS2B02G584700*), located on chromosome 2B, is characterized by increased expression in the flag leaf, which is the main assimilate source for pouring grain [69,70]. The two genes found in the *T. dicoccoides* species were not found in the *T. aestivum* genome. These are *TRIDC2AG018450* and *TRIDC2AG018420,* which encode Pyrroline-5-carboxylate reductase, P5CR, and are involved in proline synthesis. On the other hand, *T. aestivum* and *T. turgidum* have the *TraesCS2A02G147000/TRITD2Av1G042510* genes that are absent in *T. dicoccoides*. However, little is known about the function of these genes.

There are single studies devoted to *T. timopheevii* and the wild ancestral form of timopheevii lineage *T. araraticum* that concern the issue of their grain quality. These studies mainly investigate the allelic composition of gliadins and glutenins and the content of micronutrients such as iron and zinc (see review [24]). According to some data, *T. timopheevii* and the introgression lines obtained with its participation are characterized by a higher content of protein, gluten, and minerals if compared to durum wheat [40,71,72]. However, the data on the QTLs inherited from *T. timopheevii* that contributed to the quality traits are very limited. Only Shchukina et al. [38], who studied an introgression line with translocation from *T. timopheevii* in chromosome 2A, showed that the line differed by increased GC from the original bread wheat variety.

Thus, the results of the GWAS we performed suggest that the QTL on chromosome 2A is a new locus that originated from the *T. timopheevii* genome because its localization differs from the previously reported GC-associated genes/QTLs.

### 3.2. Allelic Composition of HMW Glutenins in Wheat Varieties and Introgression Lines

The allelic composition of HMW-GS was studied in spring and winter wheat varieties of Russian breeding, developed mainly in the former Soviet Union [12,73]. Comparing our results against the previously obtained data on the allelic composition of HMW-GS for the 30 samples included in this collection showed a match for 16 varieties. Differences in the allelic composition in other genotypes mainly relate to the *Glu-B1* loci. In addition, previous studies noted the seeds’ high heterogeneity [74,75], whereas, in our set, only one variety was heterogeneous at the *Glu-A1* locus (Appendix A). The differences in allelic composition found in different studies may be associated both with the heterogeneity of the seeds and their origination from different Genebanks and breeding centers, as well as with the methods for identifying the allelic composition of a locus (electrophoresis in PAAG or PCR). In our study, to identify the *Glu-1* alleles, we used the PCR markers developed for *T. aestivum*, which made it possible to differentiate most of the HMW-GS alleles. However, we failed to determine exactly which of the three By subunits (8*, 18, or 15) was present in the genome of 19 wheat varieties and seven introgression lines.

For *T. timopheevii* (genome A^t^A^t^GG) and *T. kiharae* (A^t^A^t^GGDD), no amplification products with primers for the By subunit were detected. According to Wan et al. [35], neither the 1Gx nor the 1Gy HMW subunit genes from *T. timopheevii* were amplified, which suggests that their DNA sequences are sufficiently different from the primer sequences of common wheat used for the PCR reaction. The gene encoding the new y-type HMW-GS (Gy7*) from the *T. timopheevii* G-genome was isolated by Li et al. [76]. Phylogenetic analysis showed that the new gene was closely related to the HMW-GS By (from *T. aestivum*) and Sy (from *Aegilops speltoides*) subgroups.

The commercial varieties of the collection investigated in our study had the HMW-GS set consisting mainly of the Ax1/Ax2 (90%), Bx7* + By9 (82%), and Dx5 + Dx10 (55%) subunits that determined their high baking quality (Appendix A). Most introgression lines inherited the *Glu-B1* and *Glu-D1* loci from the original wheat varieties, except for three ILs (157, 190/5-3, and 190/6-1). In line 157 (cv. Skala/*T. timopeevii*), the *Glu-A1* and *Glu-B1* loci did not correspond to the original parental forms, which suggests they possibly originated from another wheat variety. The two other ILs obtained with the participation of Chinese Spring and *T. durum* showed differences in the lengths of PCR products for the Bx7 subunit that were not characteristic of bread wheat.

A high occurrence frequency of the Ax0 subunit (45%) was also noted in the lines obtained with the participation of *T. timopheevii*. Unlike hexaploid wheat, whose genome has no Ay subunit expressed, its diploid and tetraploid wild relatives *T. urartu*, *T. boeoticum*, *T. dicoccoides*, *T. araraticum*, *T. timopheevii* are characterized by diversity in the 1A subunit [22]. However, in our study, the presence of the 1Ay subunit in the introgression lines and wheat relatives was not verified. Comparison of GC with different allelic compositions of HMW-GS loci showed that the presence of the Ax0 subunit correlates with an increase in gluten. However, GWAS did not detect significant MTAs on chromosome 1A.

## 4. Materials and Methods

### 4.1. Plant Materials and Phenotyping

To perform the investigation, a collection of 133 bread wheat varieties (*T. aestivum* L.), consisting of spring cultivars, breeding lines, and introgression lines (ILs) with substitutions and translocations from species of tribe Triticeae (*T. durum*, *T. dicoccum*, *T. dicoccoides*, *T. timopheevii*) was used. Parental wheat relatives *T. dicoccum* (AABB genome), *T. dicoccoides* (AABB), *T. timopheevii* (A^t^A^t^GG genome), and synthetic wheat *T. kiharae* (A^t^A^t^GGDD) were added to the analysis. The original accession of *T. durum* was used for the development of *T. aestivum/T. durum* ILs were not known and not included in the experimental panel. A list of plant materials is presented in Appendix A; for more detailed information on the origin of the accessions, see Leonova et al. [77]. The chromosomal localization of alien genetic material in the wheat varieties and ILs was determined based on previous karyotyping by C-banding and genotyping with SSR [78,79] and SNP markers in the framework performed in this study.

The samples were grown in the experimental field of the Institute of Cytology and Genetics, Siberian Branch of the Russian Academy of Sciences (Novosibirsk Region, 54.9191° N, 82.9903° E) in the years 2018 and 2019 sown in a randomized block design in two replicates on plots of 1 m wide, 80 grains per row and a between-row spacing of 25 cm. Sowing was carried out in the second half of May for the grain to be harvested in September during its full ripeness. The harvested grain was brought to standard moisture content (14%). The soil cover of the field consisted of leached chernozem with a humus thickness varying within 40–60 cm and a humus content of 4.2%. The soil reaction was slightly acidic (pH 6.7–6.8). The nitrogen, gross phosphorus, and potassium contents were 0.34%, 0.30%, and 0.13%, respectively.

The growing season 2018 was characterized by low temperatures in May (on average, 5 °C below normal) and high water saturation in May–June. The precipitation was 380.3 and 194.7 mm during the growing seasons of 2018 and 2019, respectively, the long-term average being 220.0 mm. The weather conditions 2019 were unstable due to uneven precipitation and temperature fluctuations in the second half of the growing season. Rainy weather was observed in May and July, and a slight drought in June and August.

Gluten content (%) was determined using an infrared express analyzer, OmegAnalyzer G (Bruins Instruments, Weiler bei Bingen, Germany). The analysis was carried out in triplicate using 8–10 g of seeds per the manufacturer’s instructions.

### 4.2. Statistical Analysis

The statistical analyses were performed using R. The significance of differences between the mean values of the two sample sets was determined using Student’s *t*-test. Broad-sense-heritability was estimated as *H^2^* = σ_g_/(σ_g_ + σ_e_/n), where σ_g_ is a genotypic variance, σ_e_ is an error variance, and n is the number of environments. Two-way ANOVA (analysis of variance) was performed to determine the significance of differences among the genotypes and the environments. The best linear unbiased estimates (BLUEs) for each accession were used, assuming the genotype as a fixed effect and the growing season as a random effect.

### 4.3. Genotyping

The DNA was extracted following the modified sodium bisulfite protocol described by [80]. DNA purification for SNP genotyping was performed using a Bio-Silica Kit for DNA Purification from Reaction Mixtures per the manufacturer’s protocol. The DNAs were then quantified using the Qubit dsDNA BR Assay kits (Thermo Fisher Scientific, Waltham, MA, USA) on a Qubit 4 Fluorometer (Thermo Fisher Scientific). SNP genotyping was performed using an Illumina Infinium 15K Wheat Platform by the TraitGenetics Section of SGS Institute Fresenius GmbH (Gatersleben, Germany, www.sgs-institut-fresenius.de, accessed on 1 August 2023). The total number of SNP markers was 13007.

### 4.4. Identification of HMW-SG Alleles

Previously reported PCR markers developed for *Glu-A1*, *Glu-B1,* and *Glu-D1* loci [45,46,47,48] were used to identify the HMW-GS alleles (Appendix A). The PCR was performed in a reaction volume of 20 μL containing 1 unit of HS-Taq PCR-Color (Biolabmix, Novosibirsk, Russia), 0.25 mM of each primer, 20 ng of genomic DNA, and sterile water. PCR products were analyzed by electrophoresis in 2% agarose gel using TBE buffer. Wheat variety Chinese Spring was used as a control.

### 4.5. Population Structure and Genome-Wide Association Study

To analyze the population structure, the R package LEA based on a STRUCTURE-like inference algorithm was used [49]. We tested 15 ancestral populations (K = 1−15) by applying the function “snmf” to select their optimal number. The calculations involved one hundred repetitions for each K, and the best replication that accounted for the lowest cross-entropy value was selected for further analysis and visualization.

The GWAS was executed using fixed and random model Circulating Probability Unification (FarmCPU) implemented in the GAPIT v.3 software R package [81] with population structure (Q matrix) as a covariate. The FarmCPU was developed with the capacity to control false positives without false negatives being compromised [50].

To search for marker-trait associations, 10,370 polymorphic SNP markers were investigated. The markers with a minor allele frequency (MAF) of less than 5% and missing data of more than 5% were excluded from the genotype dataset. The chromosome location of the SNPs was assessed using International Wheat Genome Sequencing Consortium (IWGSC) RefSeq v.1.1 and RefSeq v.2.1 submitted in the GrainGenes genomic browser [82] or the blast analysis using Ensembl Plants v. 110 [83] and the consensus genetic linkage map [51].

The quantile–quantile (Q-Q) and Manhattan plots were generated using the R package CMplot [84]. To identify significant marker-trait associations, a *p*-value threshold of 0.001 was used after applying the false discovery rate (FDR) < 0.05 correction. Linkage disequilibrium (LD) between SNP markers was calculated using the R package Genetics [85]. LD decay plots were generated using the R package LDheatmap [86].

### 4.6. Gene Annotation

To prioritize the genes linked to identified MTAs and loci, their functions were annotated using the genomic browser Persephone [87]. To assess the gene expression, Wheat Expression Browser expVIP [88] was used to select the developmental time course of the common wheat cultivar Azhurnaya [89] as a dataset. We expanded the region on the chromosome for candidate gene analysis by 100 kb to capture only genes closely linked to the significant markers that passed the FDR threshold and located within the LD interval. We decided not to expand the search area too much so as not to increase the risk of obtaining false positive candidate genes. To compare the sequences of the chromosome 2A locus in *T. aestivum*, *T. dicoccoides*, *T. turgidum*, and *T. urartu*, Ensembl Plants [83] was employed.

## 5. Conclusions

Wild and cultivated relatives of bread wheat are a reservoir of agronomically important genes that can increase the nutritional value of grain and improve the technological properties of flour products. In the present study, a GWAS was carried out to identify the genetic factors contributing to GC in grain. Using a collection of bread wheat genotypes, its relatives, and the introgression lines obtained with the participation of the representatives of the Triticeae tribe, a new locus inherited from chromosome 2A^t^ of *T. timopheevii* has been identified. It has been found that the GC in the introgression lines containing the new locus is significantly higher compared to the other wheat varieties. Further investigation will involve the development of donor lines that are close to isogenic and contain a minimum number of introgressed fragments combined with the target locus. Efforts will also be aimed at assessing the impact the new locus has on the agronomic traits and technological parameters that determine the baking quality of flour.

## Figures and Tables

**Figure 1 ijms-24-13304-f001:**
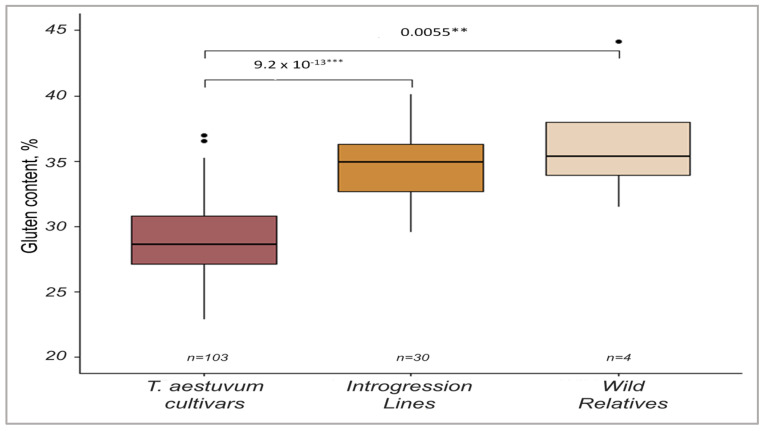
Frequency distribution of GC in spring wheat varieties, introgression lines, and wild relatives across two environments (2018−2019). “*n*” denotes the number of genotypes in each group. ** *p* < 0.01, *** *p* < 0.001. Black dot means point of outliers.

**Figure 2 ijms-24-13304-f002:**
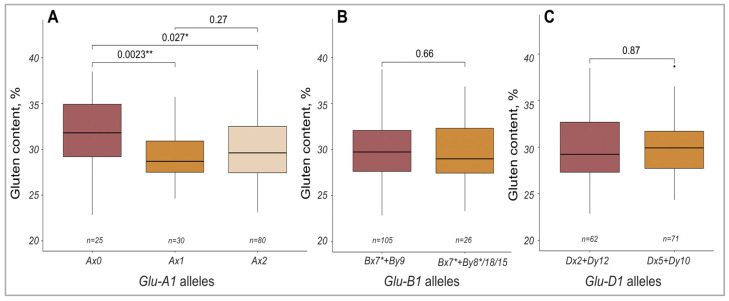
GC variability in spring wheat varieties depending on the allelic composition of the *Glu-A1* (**A**), *Glu-B1* (**B**), and *Glu-D1* (**C**) loci. “*n*” denotes the number of genotypes in each group. * *p* < 0.05, ** *p* < 0.01. Black dot means point of outliers.

**Figure 3 ijms-24-13304-f003:**
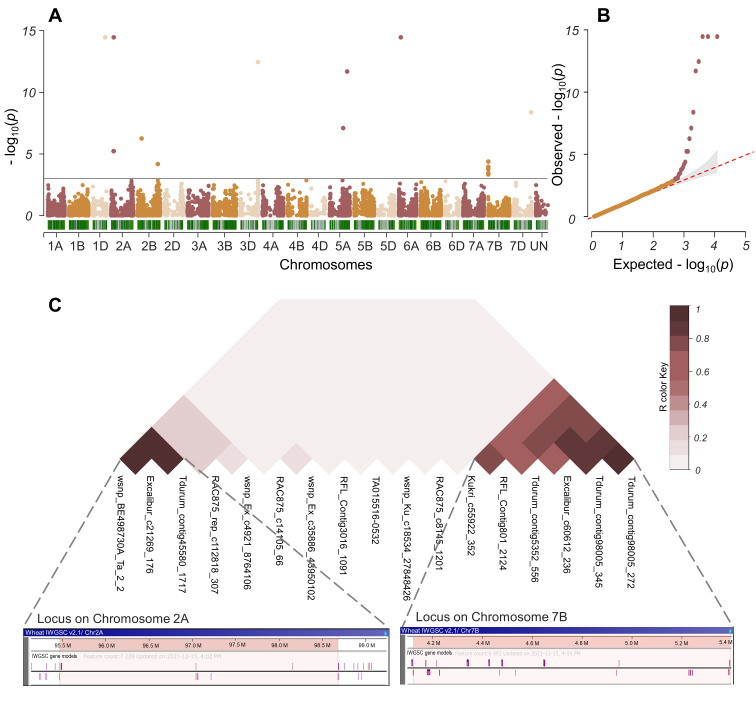
Manhattan (**A**), Q−Q (**B**), and LD plots (**C**) of genome-wide association study for GC in wheat accessions and ILs. The horizontal line indicates the threshold of significance.

**Figure 4 ijms-24-13304-f004:**
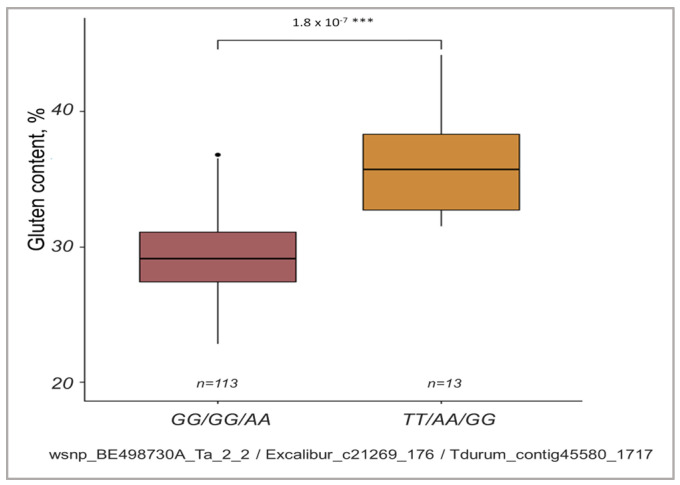
Variability of GC in spring wheat varieties depending on the haplotype composition of the quantitative trait loci on chromosome 2A. “*n*” denotes the number of genotypes in each group. *** *p* < 0.001. Black dot means point of outliers.

**Table 1 ijms-24-13304-t001:** Significant SNP markers associated with gluten content in a collection of common wheat varieties, ILs, and wheat relatives.

SNP	Chr	RefSeq v. 1.1, bp	RefSeq v. 2.1, bp	Consensus Genetic Map, cM	*p*-Value	Alleles	R^2^, %	Effect, %
wsnp_Ex_c35886_43950102	1D	436,191,827	438,667,832	115.62	3.51 × 10^−15^	C/T	0.58	0.84
wsnp_BE498730A_Ta_2_2	2A	90,741,928	95,472,680	101.97	5.83 × 10^−6^	T/G	12.32	1.78
Tdurum_contig45580_1717	2A	93,885,887	98,640,570	101.97	5.83 × 10^−6^	A/G	12.32	1.78
Excalibur_c21269_176	2A	93,926,817	98,681,112	101.97	3.51 × 10^−15^	G/A	12.32	1.78
RAC875_c14105_66	2B	146,339,548	154,587,850	unmap	5.57 × 10^−7^	C/T	1.92	0.79
RFL_Contig3016_1091	2B	711,726,897	720,190,099	unmap	6.58 × 10^−5^	T/G	2.23	−0.78
TA015516-0532	3D	609,083,324	613,195,532	unmap	3.51 × 10^−13^	G/A	8.10	1.39
wsnp_Ex_c4921_8764106	5A	474,541,596	475,037,107	53.46	7.93 × 10^−8^	C/G	11.64	−1.00
RAC875_rep_c112818_307	5A	613,477,739	615,212,548	98.90	2.05 × 10^−12^	G/A	21.15	−1.11
wsnp_Ku_c18534_27848426	6A	70,421,823	73,308,490	71.72	3.51 × 10^−15^	T/C	4.36	−1.21
Kukri_c55922_352	7B	3,796,240	4,123,555	unmap	1.55 × 10^−4^	G/A	3.81	−0.69
RFL_Contig801_2124	7B	3,848,916	4,175,664	10.06	1.73 × 10^−4^	T/C	9.18	0.77
Tdurum_contig98005_345	7B	4,907,955	5,241,285	unmap	3.61 × 10^−4^	G/A	8.21	0.72
Tdurum_contig98005_272	7B	4,908,028	5,241,358	10.06	4.56 × 10^−4^	G/A	4.59	0.76
Excalibur_c60612_236	7B	5,056,454	5,389,548	unmap	1.11 × 10^−4^	C/T	4.94	0.73
Tdurum_contig5352_556	7B	5,061,909	5,395,003	10.06	4.15 × 10^−5^	T/C	5.79	0.76
RAC875_c8145_1201	7D	610,823,574	613,256,134	133.59	4.18 × 10^−9^	A/G	8.57	−1.83

## Data Availability

All the data presented in this study are included in the manuscript and Appendix A.

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
