# Peer review of "Novel Genetic Loci from Triticum timopheevii Associated with Gluten Content Revealed by GWAS in Wheat Breeding Lines"

_ijms, 2023, doi:10.3390/ijms241713304_

Round 1
Reviewer 1 Report
The manuscript is a very good attempt on identification of loci for quality traits in wheat and holds immense importance for grain quality breeding in wheat. However, addressing the suggestions will further improve its quality and bring clarity to readers.
# Authors did not mention about the False discovery rate or Bonferroni Correction Threshold for QTN significance. Please add this in the relevant section.
# Line 22: "we supposed that the locus on chromosome 2A inherited from T. timopheevii was potentially novel"
instead of supposed use "inferred" or "concluded" and replace was with "is".
#Line 45- high molecular weight subunits (HMW-GS) should be "high molecular weight glutenin subunits (HMW-GS)".
#Line 74- as SDS sedimentation.. expand SDS
"Line 76- For the species with genomes G, S, E, J, and others.. not clear which genomes the authors are referring to. Please make it clear by re-writing and detailing about the genomes.
# Line 88- "currently practically absent" change to "currently lack".
# Line 97- Subheading- 2.1. Phenotypic assessment.. After Table S1... authors directly jump to TableS3.. Why Table S2 skipped and occurs in next paragraph? Please resequence the tables.
High molecular weight and gluten content has been abbreviated in Introduction, why authors keep on using expanded form later in different sections? Please abbreviate at all places.
# Line 226- the analyzed region was expanded by ~100 Kb 226 in both directions from the extreme markers. Why only 100kb was chosen? Please make clear the criterion for choosing ~100kb.
# Line 337-340- "However, the data on the QTLs inherited from T. timopheeviithat contributed to the quality traits is absent. Only Shchukina et al. [38] who studied an introgression line with a translocation from T. timopheeviiin chromosome 2A showed that the line differed by in-creased GC from the original bread wheat variety.
change absent to "very limited" or "lack".
Please correct the highlighted texts in the attached PDF.

Moderate editing of English language required for grammatical mistakes and typographical errors
Author Response
Dear Reviewer,
The authors are grateful to Reviewer for a careful reading of the manuscript, valuable comments and English correction. In general, we agree with all the reviewers' suggestions for improving the manuscript.
Below are answers to the comments:
Reviewer 1.
- # Authors did not mention about the False discovery rate or Bonferroni Correction Threshold for QTN significance. Please add this in the relevant section.
Answer: False discovery rate was added in Material and Methods (page , line 466-468)
2. # Line 22: "we supposed that the locus on chromosome 2A inherited from T. timopheevii was potentially novel" instead of supposed use "inferred" or "concluded" and replace was with "is".
Answer: corrected
- #Line 45- high molecular weight subunits (HMW-GS) should be "high molecular weight glutenin subunits (HMW-GS)".
Answer: corrected
- #Line 74- as SDS sedimentation.. expand SDS
Answer: corrected
- "Line 76- For the species with genomes G, S, E, J, and others.. not clear which genomes the authors are referring to. Please make it clear by re-writing and detailing about the genomes.
Answer: paragraph re-written (page 2, lines 77-86)
- # Line 88- "currently practically absent" change to "currently lack".
Answer: corrected
- # Line 97- Subheading- 2.1. Phenotypic assessment.. After Table S1... authors directly jump to TableS3.. Why Table S2 skipped and occurs in next paragraph? Please resequence the tables.
Answer: table numbering corrected
- High molecular weight and gluten content has been abbreviated in Introduction, why authors keep on using expanded form later in different sections? Please abbreviate at all places.
Answer: corrected
- # Line 226- the analyzed region was expanded by ~100 Kb 226 in both directions from the extreme markers. Why only 100kb was chosen? Please make clear the criterion for choosing ~100kb.
Answer: A physical map of the pseudomolecule (RefSeq v. 1.1 and RefSeq) was used to identify candidate genes. However, the exact position of candidate genes may differ in the experimental population. That's why we expanded the region on the chromosome for candidate gene analysis by 100 kb to capture genes only closely linked to the significant markers which passed the FDR threshold and located within the LD interval. The annotated genes within 250 Kb of the mapped SNP were considered candidate genes as described in other papers (see Ahmed et al. Agriculture 2020, 10, 392; doi:10.3390/agriculture10090392). We decided not to expand the search area too much, so as not to increase the risk of obtaining false positive candidate genes.
Clarifications was added to Materials and Methods (page11, line 475-479)
- # Line 337-340- "However, the data on the QTLs inherited from T. timopheevii that contributed to the quality traits is absent. Only Shchukina et al. [38] who studied an introgression line with a translocation from T. timopheevii in chromosome 2A showed that the line differed by in-creased GC from the original bread wheat variety
change absent to "very limited" or "lack".
Answer: change on “very limited”
- Please correct the highlighted texts in the attached PDF.
Answer: highlighted text in the PDF file was corrected
Thank you very much
With kind regards
Dr. Irina Leonova
leonova@bionet.nsc.ru
Reviewer 2 Report
Review of the manuscript titled: Novel genetic loci from Triticum timopheevii associated with 2 gluten content revealed by GWAS in wheat breeding lines.
The study is devoted to an important subject, uses unique material, relevant methodology and arrives to important results and conclusions. It is certainly significant contribution to wheat science and deserves publication. However, as any manuscript it can be improved by attending a few issues.
1. Lines 98-107. The reference to supplemental tables is not in order: Tables S1, then S3 and S4 – where is Table S2.
2. Figures 1, 2, 5. It would be useful to mention how many genotypes in each group presented in the figures.
3. Table 1. Last column Effect – units need to be specified.
4. It would be useful to add a table in the body of the paper with the list of genotypes with highest gluten content, say top 15-20, their origin and presence/absence of the key SNPs. Cultivar Rassvet (top 8) is the only in this group which does not possess any introgression and would be interesting to know if it possesses any useful SNPs.
5. The main deficiency of the study is that gluten content is taken out of the context of protein content. Frequently, the two traits are closely associated. The authors are encouraged to analyze the relationship between the two traits. If there are large differences between the genotypes and groups for protein content and it is closely associated with the gluten content, perhaps the authors need to adjust gluten content to uniform protein content based on regression. Then they can use adjusted gluten content values to run GWAS and see if the results using original and adjusted values match. May be some other significant SNP will be identified and the ones significant will not be proven. The focus can be on significant SNPs using original and adjusted values.
Author Response
Dear Reviewer,
The authors are grateful to Reviewer for a careful reading of the manuscript and valuable comments. In general, we agree with all the reviewers' suggestions for improving the manuscript.
Below are answers to the comments:
- Lines 98-107. The reference to supplemental tables is not in order: Tables S1, then S3 and S4 – where is Table S2.
Answer: table numbering corrected
- Figures 1, 2, 5. It would be useful to mention how many genotypes in each group presented in the figures.
Answer: In Figures 1, 2, 5, at the bottom of each group, the number of samples for each genotype is indicated. For more clarity, explanations have been added to the figures’ captures «“n” denotes the number of genotypes in each group
- Table 1. Last column Effect – units need to be specified.
Answer: units added
- It would be useful to add a table in the body of the paper with the list of genotypes with highest gluten content, say top 15-20, their origin and presence/absence of the key SNPs. Cultivar Rassvet (top 8) is the only in this group which does not possess any introgression and would be interesting to know if it possesses any useful SNPs.
Answer: thank you very much for your recommendation. We have prepared this table, but we suggest to place it in Supplementary material as Table S7. See table description on page7, lines 222-229
- The main deficiency of the study is that gluten content is taken out of the context of protein content. Frequently, the two traits are closely associated. The authors are encouraged to analyze the relationship between the two traits. If there are large differences between the genotypes and groups for protein content and it is closely associated with the gluten content, perhaps the authors need to adjust gluten content to uniform protein content based on regression. Then they can use adjusted gluten content values to run GWAS and see if the results using original and adjusted values match. May be some other significant SNP will be identified and the ones significant will not be proven. The focus can be on significant SNPs using original and adjusted values.
Answer: thank you very much for your recommendation. Unfortunately, in this article we will not be able to make correlations and analyzed relationship between grain protein content and gluten content, since there are no data on the protein content for the entire collection (especially for introgression lines). Previously, we performed GWAS for the protein content of common wheat varieties that are present in the experimental panel used in this work. For common wheat, it was shown that one of the QTL mapped on chromosome 6A was associated with both protein and gluten and gluten contents. We discussed this data in the Discussion section (page 8, lines 290-295).
However, we will take your recommendation into account in our future work.
With kind regards
Dr. Irina Leonova
leonova@bionet.nsc.ru